# Transformation of temporal sequences in the zebra finch auditory system

**Yoonseob Lim[1,2], Ryan Lagoy[3], Barbara G Shinn-Cunningham[4], Timothy J Gardner[4,5]***

[1]Department of Cognitive and Neural Systems, Boston University, Boston, United States; [2]Convergence Research Center for Diagnosis, Treatment, and Care System for Dementia, Korea Institute of Science and Technology, Seoul, Korea; [3]Department of Electrical and Computer Engineering, Boston University, Boston, United States; [4]Department of Biomedical Engineering, Boston University, Boston, United States; [5]Department of Biology, Boston University, Boston, United States

**Abstract** This study examines how temporally patterned stimuli are transformed as they propagate from primary to secondary zones in the thalamorecipient auditory pallium in zebra finches. Using a new class of synthetic click stimuli, we find a robust mapping from temporal sequences in the primary zone to distinct population vectors in secondary auditory areas. We tested whether songbirds could discriminate synthetic click sequences in an operant setup and found that a robust behavioral discrimination is present for click sequences composed of intervals ranging from 11 ms to 40 ms, but breaks down for stimuli composed of longer inter-click intervals. This work suggests that the analog of the songbird auditory cortex transforms temporal patterns to sequence-selective population responses or 'spatial codes', and that these distinct population responses contribute to behavioral discrimination of temporally complex sounds.

*For correspondence: timothyg@ bu.edu

## Introduction

A highly developed auditory network supports auditory-vocal behavior in songbirds. The core of the auditory processing system consists of anatomical areas named Field L, NCM (caudomedial nidopallium), and CM (caudomedial mesopalium) (*Vates et al., 1996*) (*Figure 1c*). These areas and other associated auditory areas are directly and indirectly connected with the song motor pathway (*Vates et al., 1996*; *Mandelblat-Cerf et al., 2013*). Field L, the primary thalamorecipient area, is composed of four different sub-regions (L2a, L2b, L1, and L3) that are reciprocally connected (*Vates et al., 1996*). Among these sub-regions, L2a receives the strongest input from the core of Ov (nucleus ovoidalis), the primary auditory thalamus (*Müller and Leppelsack, 1985*; *Rübsamen and Dörrscheidt, 1986*; *Hose et al., 1987*). Secondary auditory areas — L2b, L3, and L1 — receive feedforward input from L2a and thalamus, but also receive feedback from higher cortical areas such as CM. These hierarchically and reciprocally connected auditory areas are thought to be analogous to the early stages of mammalian auditory cortex, but the details of the homologies remain a subject of debate (*Jarvis et al., 2005*; *Wang et al., 2010*; *Calabrese and Woolley, 2015*).

For zebra finches and other songbirds, temporal cues in song provide reliable information about the identity of the singer and are used for perceptual discrimination of songs (*Gentner and Margoliash, 2003*; *Gentner et al., 2006*; *Grace et al., 2003*; *Shaevitz and Theunissen, 2007*). The songbird auditory processing stream is well adapted to this information-processing task and reliably relays temporal information in conspecific song. In the zebra finch auditory system, there are neurons from midbrain to the highest levels of auditory association areas that respond with precise spike times to playback of conspecific song. This is true for both dense-spiking neurons and the highly

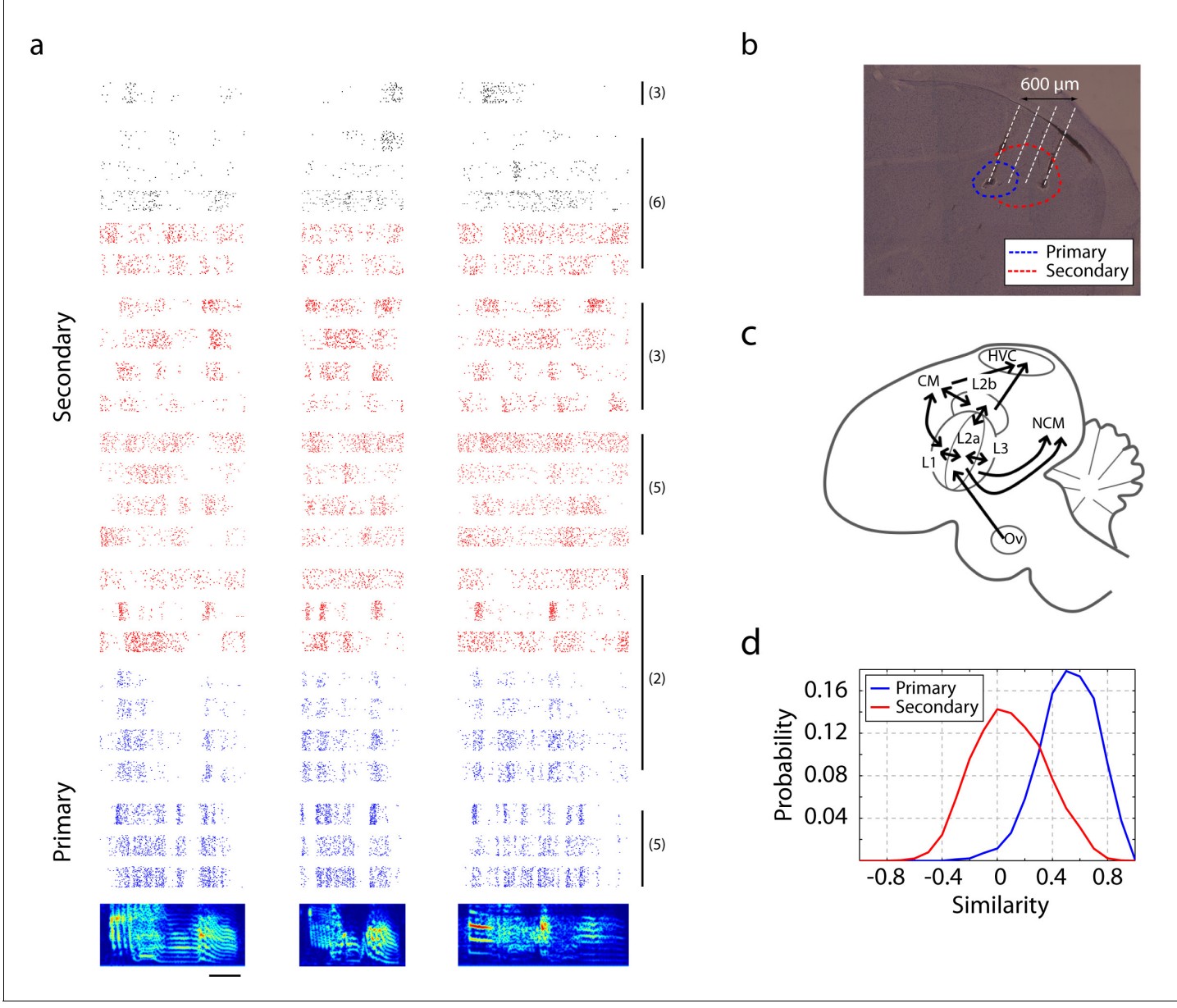

**Figure 1.** Neural responses in primary and secondary auditory areas to birdsongs. (a) Example of neural responses in primary (blue) and secondary auditory areas (red and black) to birdsongs. Syllable responses were extracted from playback of whole songs. Individual cells in this figure were recorded in different birds. Numbers on the right correspond to the bird indices shown in *Figure 1—figure supplement 1*. Cells in the primary auditory area, L2a, respond more synchronously than cells in the secondary area. Red and black colors in the raster denote two classes of cells in secondary auditory areas defined by spike-width. (For red, spike width is less than 250 μs , and for black, greater than 250 μs.) The scale bar is 50 ms. (b) Sagittal section located at 1.5 mm lateral of the midline with estimated electrode shank positions (dotted white line). Physiological locations are confirmed by the anatomy (*Figure 1—figure supplement 1*). (c) Schematic of a sagittal section of male zebra finch brain. (d) Response similarity scores between all pairs of cells in the secondary auditory area are lower than similarity scores in the primary auditory area. (Secondary auditory responses to song are more diverse across neurons.)

The following figure supplements are available for figure 1:

**Figure supplement 1.** Estimated recording location of units.

**Figure supplement 2.** Song syllable discriminability analysis.

**Figure supplement 3.** Peri-stimulus time histogram (PSTH) of song responses.

selective, sparse-firing neurons recently described in the high-level auditory area, NCM (*Schneider and Woolley, 2013*), as well as in the auditory-motor association area, HVC (high vocal center) (*Prather et al., 2008*). Using a spectrotemporal receptive field (STRF) analysis, the effective temporal integration window of neurons in L2a, the first thalamorecipient zone, was observed to be very brief compared with responses one step further from the periphery in areas L1 and L3 (*Kim and Doupe, 2011*). The secondary areas (including L2b, L1, and L3) but not the primary zone (L2a) are recipients of significant feedback from high-order auditory areas (*Vates et al., 1996*). In combination, the results of these few studies suggest that an interesting transformation of temporal sequences could take place between primary and secondary zones in Field L.

Here, we developed new experimental paradigms to examine how temporally patterned auditory stimuli are transformed in the transition from the primary thalamorecipient zone, L2a, to the secondary auditory processing areas, L2b and L3. We first demonstrate that neurons responding to song in secondary areas, L2b and L3, become less synchronous in their relative response times yet more informative about the identity of specific syllables when compared to those in the primary area, L2a.

To zero in more closely on the nature of the transformation, we examined responses to a set of simplified auditory stimuli consisting of click sequences. The chosen stimuli were akin to 'Morse code' — the sounds differed only in the temporal ordering of intervals between clicks. These intervals were drawn from a distribution similar to the intervals between sub-syllabic acoustic transitions in zebra finch song (*Gardner et al., 2001*; *Amador et al., 2013*). For click sequence listening, a distinctive transformation of auditory responses was found between primary and secondary auditory zones. In the primary zone, each click elicited a similar low-latency response in all recorded neurons, and the structure of this response was largely insensitive to the temporal context of the click. One synapse further from the periphery, in secondary auditory areas, L2b and L3, neurons responded asynchronously and selectively, depending on the temporal context of the click. In effect, temporal sequences are transformed to distinct population vectors in the transition from primary to secondary auditory areas. In this process, temporal patterns come to be represented in a format that could directly form the basis of perceptual discriminations based on simple thresholds.

We next tested whether songbirds could discriminate different temporal click sequence patterns in an operant-training paradigm. A novel 'restart-go' operant paradigm, which we found effective for particularly challenging discrimination tests in zebra finches, was developed for this purpose. Using this training procedure, zebra finches rapidly learned to discriminate click sequences that were composed of song-like intervals. When the stimulus set was slowed by a factor of two, the strength of the temporal to spatial transformation in the secondary auditory was reduced, and there was a corresponding degradation of behavioral discrimination.

Taken together, these results indicate that the ascending auditory pathway in zebra finches transforms temporal sequences into distinct population vectors. This transformation applied to click sequences consisting of intervals that overlap with sub-syllabic acoustic structure in song, and may provide an important substrate for song perception and discrimination in sub-syllabic time-scales.

## Results

General note: the electrophysiological recordings reported here were gathered using four-shank, 32 channel silicon electrodes. From each bird, we recorded activity simultaneously from the primary thalamorecipient zone in the auditory area, Field L2a, and neighboring auditory areas in L2b and L3 (*Figure 1a and b*). All stimuli were presented in an interleaved fashion, and each animal was recorded acutely, with all data gathered in a single session. All data presented in figures and quantified below were gathered from well-sorted single-unit responses — a minority of recordings (*Figure 3—figure supplement 3*). The only exception to this rule is *Figure 3—figure supplement 2*, which includes a few channels of high SNR multi-unit traces that did not satisfy our criterion for single-unit isolation. These traces are marked with an asterisk. For additional details, see Methods.

### Transformation of song responses in the auditory hierarchy

We first compared the temporal coding of song in primary (L2a) and secondary auditory areas (L2b and L3) of unanesthetized songbirds. Our intent was not to thoroughly catalog song responses, but rather to calibrate responses in order to design a set of synthetic stimuli that could be used for the

remainder of the study. Primary and secondary recording sites were distinguished on the basis of the distinct response profiles found in the two areas (*Figures 1a* and *3a*). This classification was confirmed by spatial mapping of the recording sites (*Figure 1—figure supplement 1*), showing that the primary cells were spatially segregated from the secondary neurons. Due to small anatomical and surgical variations and the small scale of the primary zone, this area could not be reliably identified by spatial coordinates alone.

Precise spike timing could be found in both primary and secondary areas in response to song. Focusing first on responses in the primary auditory area, L2a, we found a surprising degree of response synchrony across neurons and across birds (*Figure 1a*). The population peri-stimulus temporal histogram (PSTH) for each song was deeply modulated for neurons in L2a (*Figure 1—figure supplement 3*, *Figure 2a* is the histogram of inter-peak intervals in this population PSTH). By contrast, neurons in secondary auditory areas, L2b and L3, showed a broader repertoire of response profiles. This increase in the diversity of response timing leads to a decrease in the magnitude of the cross-correlation between the PSTHs of individual neurons in the secondary auditory areas relative to a similar cross-correlation performed in primary area, L2a (*Figure 1d*).

## Transformation of click-sequence responses in the auditory hierarchy

Our next objective was to examine whether a similar transformation from synchronous to asynchronous coding could be seen for more elementary stimuli consisting of irregularly spaced clicks. This synthetic stimulus would allow us to probe whether the sequence transformation from the primary to the secondary auditory areas requires complex spectral content. If secondary auditory neurons have more complex or more selective spectral receptive fields, the emergence of asynchronous coding in the secondary auditory areas could be explained on the basis of this acoustic selectivity alone. However, if the transformation from synchronous primary response to asynchronous secondary responses could be reproduced with click trains, the result would indicate that the auditory processing pathways contain intrinsic temporal dynamics that transform temporal sequences independent of spectral selectivity.

The chosen synthetic stimuli were three seconds long and composed of clicks separated by ten specific inter-click intervals (11, 14, 16, 20, 23, 26, 29, 34, 36 and 40 ms). We chose these intervals on the basis of the timescale of neural responses to birdsongs in L2a (*Figure 2a and b*). The inter-peak intervals of the population PSTH in response to these click sequences was similar to inter-peak intervals in response to natural song. In effect, we chose click patterns that, in the primary auditory area, elicited a temporal response that loosely overlapped with the natural song response. We note that the selected inter-click intervals are also similar to intervals between sub-syllabic acoustic transitions found in zebra finch song (*Amador et al., 2013*; *Norton and Scharff, 2016*). For comparison, *Figure 2* also shows the L2a PSTH inter-peak interval histogram for click sequences slowed by a factor of two.

The duration of all ten click intervals summed together is 249 ms. The longer three-second

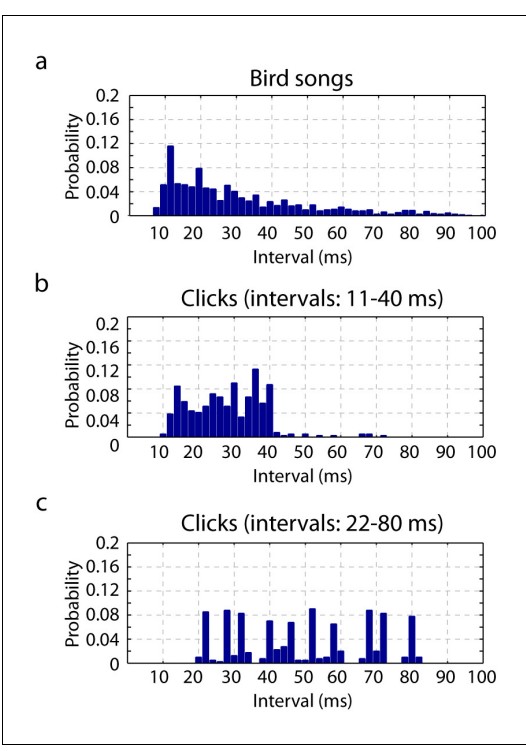

**Figure 2.** Timescales of neural responses in the primary auditory area, L2a. (a) Interval histogram of peaks in the PSTH of neurons in the primary auditory area, L2a, in response to bird songs. The population PSTH contains intervals distributed from 10 ms to 40 ms. (b,c) Interval histogram of peaks in the PSTH of neurons in the primary auditory area, L2a, in response to click sequences. For the click patterns, we applied two different timescales for the click intervals. In the first timescale, the click sequence evokes PSTH intervals in the range of 10–40 ms. The slower set of stimuli evokes PSTH intervals in the range of 20–80 ms.

sequences were built from 249 ms blocks, in which each block contains a permutation of the ten click intervals. In some stimuli, the blocks were repeating and in others non-repeating. For all sequences, the stimuli differed only in the ordering of click intervals. On timescales longer than the block duration, the statistical properties of all stimuli were equivalent. The set of stimuli used in this study can be seen in *Figure 3—figure supplement 1*. (Sample audio files are also provided. See *supplementary file 1*).

Raster plots for single units in primary and secondary auditory areas are shown in *Figure 3a*. (Example spike waveforms of single units and corresponding rasters are shown in *Figure 3—figure supplement 3*.) Raster plots for the full ensemble of single and multi-units are shown in *Figure 3—figure supplement 2*, including a breakdown of secondary cells into narrow (red) and broad-spiking (black) neuron waveforms. Only narrow units were found in the primary auditory area. (This figure is the only time in the paper that poorly sorted units, or 'multi-units' are included.) A distinct change in the temporal response to click sequences can be found in the transition from primary to secondary areas. In the primary auditory area, the click responses are fairly insensitive to the local context – to first approximation, each click evoked a synchronous, low latency response across channels, whereas secondary auditory areas were characterized by sparser and less synchronous responses that were more sensitive to the sequence context of the click (*Figure 3—figure supplements 4* and *5*). The click sequence, by definition, contains no significant spectral cues for frequencies above 100 Hz (the shortest interval in the click set was 11 ms, thus below the 100 Hz cutoff). Zebra finch hearing thresholds for pure tones are attenuated by about 20 dB relative to humans at 100 Hz (*Okanoya and Dooling, 1987*; *Moore, 2007*), and the fundamental frequency of conspecific song is typically 500 Hz or higher in zebra finches.

As for song responses, the transition from primary to secondary thalamorecipient areas reveals a desynchronizing transformation that maps temporal click sequences onto distinct neuronal ensembles. For the click sequences used here, this transformation is even more apparent than for song responses. The diversification of neuronal responses increases the information about the preceding temporal context of a given click that the population vector contains. To demonstrate this, we computed phase-space trajectories of the population vector in response to click sequences, and then quantified the Euclidean distance between these phase-space trajectories. In this analysis, every neuron recorded defines a direction in a phasespace hypercube, and the average firing rate of the cell defines a position along the respective axis.

The phase-space trajectory for three cells in the primary auditory area and three cells in the secondary auditory areas during playback of two distinct sequences are shown in *Figure 4a*. In the primary auditory area, L2a, the phase-space trajectories of distinct stimuli overlap for all time points, meaning that the pattern of active cells contains little population-vector information that can distinguish the stimuli. By contrast, in secondary auditory areas, specific points in the phase-space trajectory diverge from one another in a stimulus-dependent manner. That is, the pattern of cell responses in secondary auditory areas contains information about one or more intervals preceding the click. To summarize simply – there are particular configurations of active cells that occur only during playback of one stimulus or another — a useful feature for a system that is tuned to make fine discriminations about temporal sequence patterns.

To quantify the degree to which the click stimuli can be distinguished on the basis of the neural responses, we defined a simple decoding mechanism based on the population vector of the ensemble response (see Methods for details). In this decoding, the discriminability of the sequence at a particular time is given by the distance in phase space to the nearest trajectory belonging to a different stimulus. The power of this 'spatial' code for sequence discrimination is quantified through an ROC (receiver operating characteristic) analysis in *Figure 4b*. We analyzed coding in primary and secondary areas using the ROC analysis, using a fixed number of single unit recordings (n = 10) in both cases. In the secondary auditory areas, but not the primary thalamorecipient area, temporal sequences are mapped onto distinct population patterns, revealing a better sequence decoding in the ROC analysis (spike times of the units used in this analysis are also provided in *Figure 4—source data 1*). We repeated this analysis for just the first 500 ms of the stimulus, and still found a high degree of sequence discriminability in the secondary auditory areas (*Figure 4—figure supplement 1*). This shorter analysis is more directly relevant to the behavioral discriminations reported below, as trained birds performing behavioral discriminations typically respond within this time frame (*Figure 6—figure supplement 2*). To further validate this approach, we applied the same analysis to the

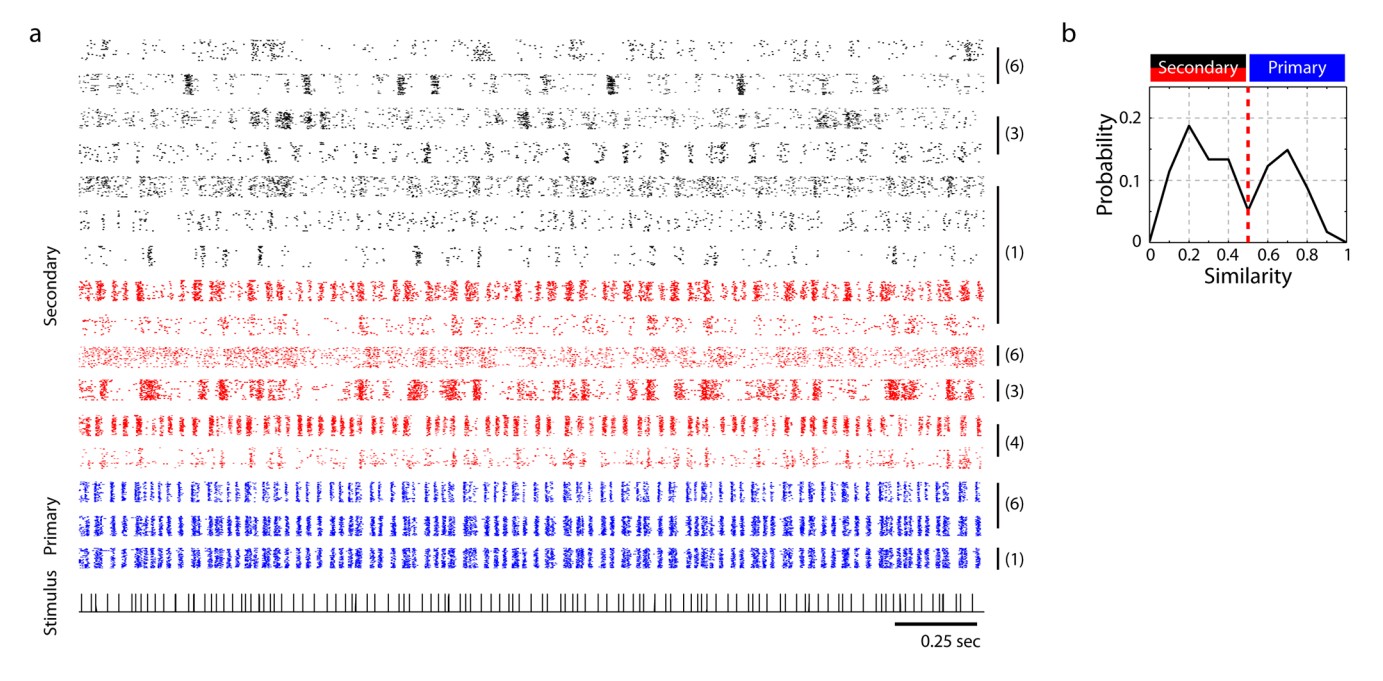

**Figure 3.** Neural responses to click sequences in primary and secondary auditory areas. (**a**) Example of neural responses in primary and secondary auditory areas. Units from individual birds are grouped (black vertical bars and corresponding bird indices are shown on the right of the rasters). Red and black rasters mark two classes of cells in secondary auditory areas that are defined by spike-width. For red rasters, the spike width is less than 250 µs and for black, greater than 250 µs. Blue rasters are cells in the primary auditory area, L2a. (**b**) Histogram of cross-correlation scores between the click stimulus and the PSTH response. The discrimination line between two peaks (at 0.5 similarity score) also segregates the cells spatially (*Figure 1— figure supplement 1*), confirming the classification of neurons as residing in spatially separated areas – either L2a or L3/L2b.

The following figure supplements are available for figure 3:

**Figure supplement 1.** All click sequences used for neural recordings and operant training.

**Figure supplement 2.** Combined single and multi-unit responses to sequence 1 and sequence 2.

**Figure supplement 3.** Example spike waveforms corresponding to click responses shown in raster form.

**Figure supplement 4.** Population PSTH of neurons in response to click sequences.

**Figure supplement 5.** Latency of neural responses to click sequences in the primary auditory area, L2a.

PSTH of the song syllable responses (n = 13 syllables, *Figure 1a*) and found an increase in syllable discriminability in the secondary auditory area (*Figure 1—figure supplement 2*). Given the rich spectral content of song relative to clicks, the primary area, L2a, already shows a high degree of response selectivity, better than that in the response to the click sequences.

We next repeated the click electrophysiology using a stimulus set composed of intervals twice as long as those in the first stimulus set (22–80 ms, rather than 11–40 ms, *Figure 2c*). This change in stimulus timescale had a minimal impact on spike rate in the secondary auditory cortex (*Figure 5— figure supplement 1*), but resulted in a significant change in the power of the temporal to spatial transformation. Using the same phase-plane ROC analysis, we found that the timescale dilation led to reduced sequence discrimination in secondary auditory areas (*Figure 5*).

## Behavioral recognition of click sequences

The preceding electrophysiology experiments demonstrated a transformation of click responses to distinct population vectors in the secondary auditory areas of unanesthetized zebra finches. As a

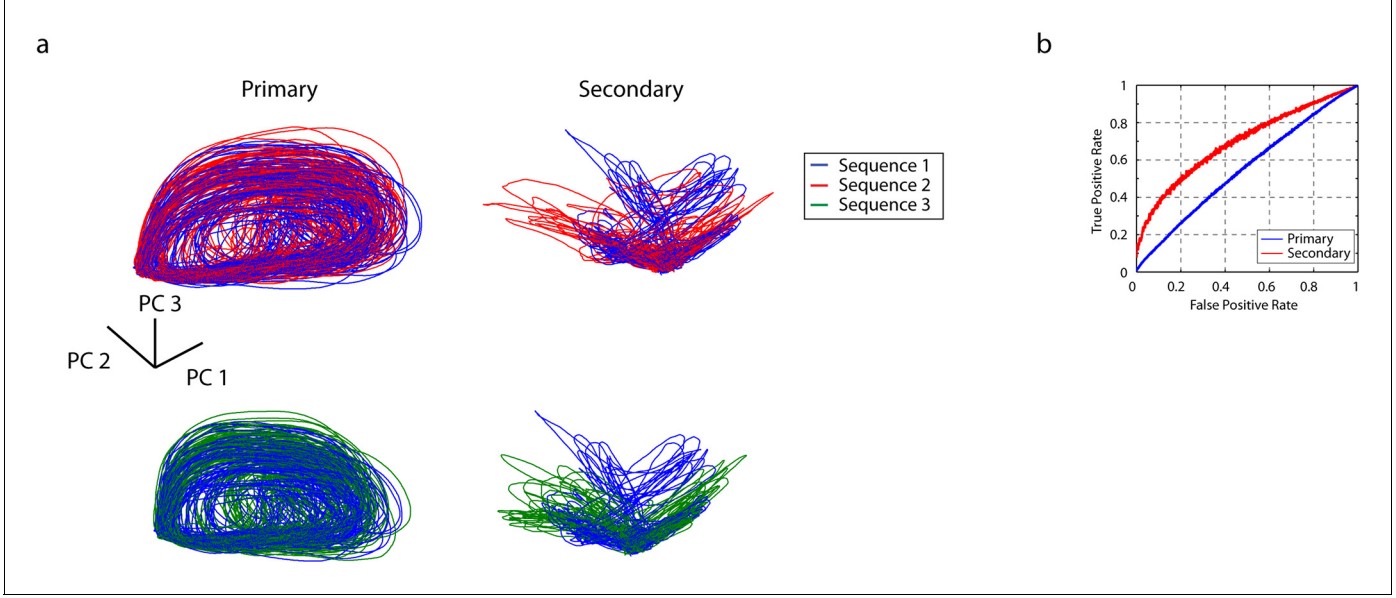

**Figure 4.** Temporal sequences are transformed to distinct population vectors in the secondary auditory areas, L2b and L3. (a) For different stimuli, ensemble state-space trajectories are discriminable in secondary auditory areas but not in the primary auditory area, L2a. For each trace, the bin size for the ensemble state space was 5 ms. Each trace is smoothed by rectangular windows (10 ms) for visualization. (b) Receiver operating characteristic (ROC) analysis reveals enhanced discriminability of click sequences in secondary auditory areas, L2b and L3, relative to those in the primary auditory area, L2a.
The following source data and figure supplement are available for figure 4:

**Source data 1.** Source data for ROC curve.
**Figure supplement 1.** Short click-sequence discriminability analysis.

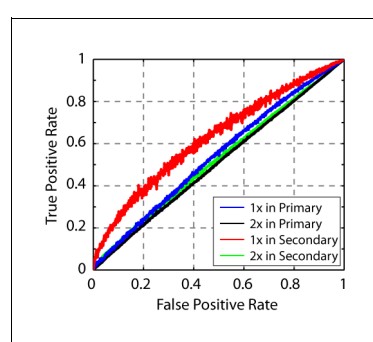

**Figure 5.** Neural sequence discriminability depends on the timescale of the click sequence. ROC analysis reveals that the discriminability of the click sequences is constrained by the interval distribution of the click stimuli. When the sequence is slowed by a factor of two, the discriminability of click sequences is lost in the secondary auditory area (shown in green).
The following figure supplement is available for figure 5:

**Figure supplement 1.** Spike rate of cells in response to click sequences with different timescales.

result, areas downstream of secondary auditory areas could, in principle, solve a click-sequence classification problem by applying a simple summation and threshold to subsets of secondary cell inputs. Given the robust transformation of temporal click sequences in zebra finch auditory areas, we next sought to determine whether songbirds could be trained to behaviorally discriminate this class of artificial stimuli, and whether or not properties of the electrophysiological responses correlated with behavioral discriminations.

Songbirds were trained using a new operant-training procedure developed for this study (***Figure 6—figure supplement 1***). We call this automated training 'reset-go'. A detailed description of the training procedure can be found in the Methods. In essence, a bird can demonstrate learning through two behaviors — by interrupting the playback of a non-rewarding stimulus to 'request' the reset of an unfavorable trial or by accessing the water port during playback of rewarding stimuli. In all experiments, two sounds were presented — a rewarded stimulus (click sequence 2 from ***Figure 3—figure supplement***

1) and non-rewarded stimulus (click sequence 1 or 9 from *Figure 3—figure supplement 1*). Zebra finches in this task were mildly water restricted, and worked for 1–5 µl drops of water, routinely performing a thousand trials in a five-hour training session.

*Figure 6a* reveals the time-course of discrimination learning for one bird. Ten days after the initiation of training, this bird would interrupt the playback of the unrewarded stimulus (sequence 1) within three seconds and access the water port while the rewarded stimulus (sequence 2) was presented. *Figure 6b* shows summary statistics for learning in eight birds trained to discriminate sequence 1 vs. sequence 2. *Figure 6—source data 1* documents the groups of birds trained and *Figure 6—figure supplement 2* shows the time-course of learning for the various groups. The detailed training procedure is described in the Methods. Within a population of trained birds (n = 53), a majority (n = 35 birds) showed high levels of performance (d' > 1) within 14 days of training onset, revealing that songbirds could readily learn to discriminate the fast temporal click sequences used in this study.

## Catch trials probe the nature of auditory discrimination

To probe the underlying nature of the auditory discrimination, we examined catch trials for two conditions. For time-reversed click stimuli, behavior fell to chance levels (*Figure 7a*), indicating that the ordering of the click intervals was critical to the behavioral discrimination. The next test examined cyclic permutations of the training stimuli. Rather than beginning playback at the normal starting interval of each sequence, the cyclic permutation initiated each stimulus at a random click interval in the three second stimulus – effectively a phase shift in the stimulus. For this group of catch trials, a

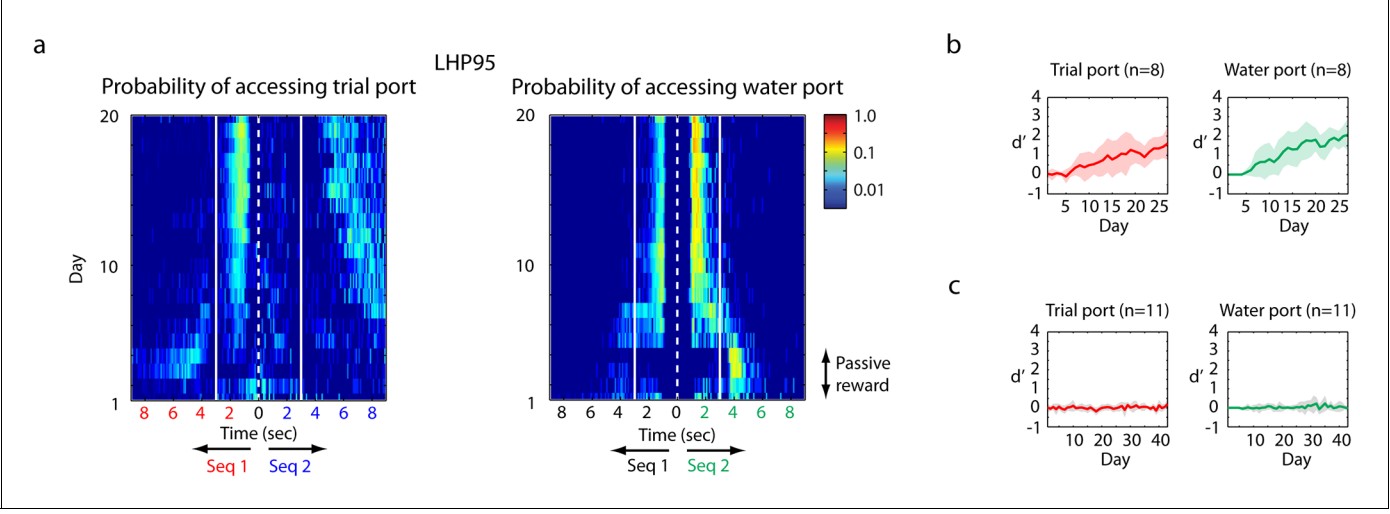

**Figure 6.** Operant training with click sequences. (a) Example of training by the single-stage behavioral-shaping method. The probability distribution of accessing trial port and water port is illustrated on a log scale. The white dotted line represents the start of sequence playback and the white solid line is the termination time of the stimulus. We show two stimuli back to back with mirrored time axes. Asymmetry between the solid lines in this image indicates learning. Over the course of training, this bird started to interrupt playback of the non-rewarding sequence by accessing the trial port before sequence 1 (the unrewarded sequence) stopped playing. The bird also learned to access the water port selectively during the playback of the rewarded sequence (sequence 2). (b) Learning curve for birds exposed to the single-stage training method (n = 8 birds). With the single-stage training method, most birds start to show differentiated responses (d' is around 1) after two weeks of training; that is, they interrupt and reset sequence 1 playback and access the water port for sequence 2 playback. (c) When the click intervals are slowed by a factor of two, all trained birds (n = 11 in the single-stage method) were unable to discriminate the temporal sequences; d' is around 0.

The following source data and figure supplements are available for figure 6:

**Source data 1.** Summary of training.

**Figure supplement 1.** Operant training setup.

**Figure supplement 2.** Result of operant training.

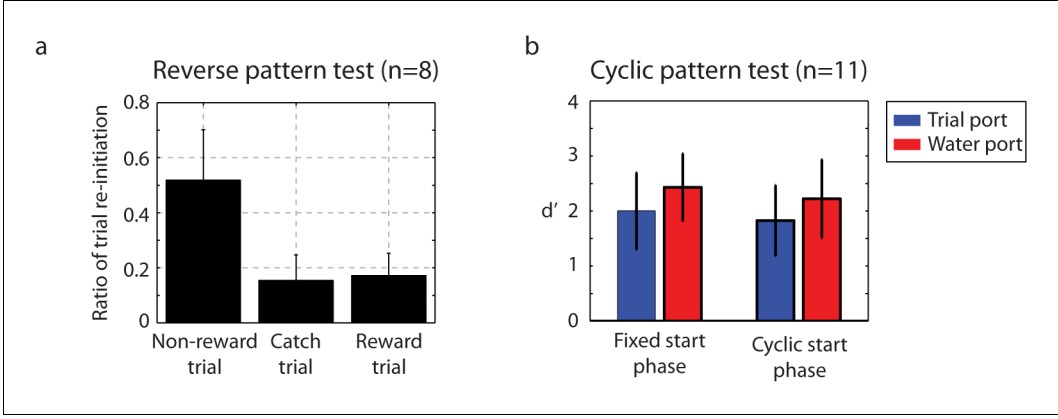

**Figure 7.** Catch-trial analysis. (**a**) During catch-trial analysis, for 10% of non-rewarding trials, we presented reverse patterns to eight birds. The birds did not show any recognition of the reverse pattern (catch trials). Only the familiar non-rewarded sequence led to the adaptive behavior of resetting playback. Mean ± s.d. of trial interruption ratio is shown. (**b**) In this cyclic permutation catch-trial analysis, playback of the click sequence started at a random interval in the repeating sequence on each trial (a phase shift in the stimulus order); all birds (n = 11) maintained performance. This indicates that discriminations were based on patterns of click intervals regardless of the absolute time of any specific click relative to trial onset.

small decrease in performance was found when the cyclic stimulus was first introduced, but within four days, performance returned to baseline (*Figure 7b*). The conclusion from this is that the birds are listening for patterned sequences of intervals irrespective of their absolute time of occurrence relative to the onset of the trial.

## Breakdown of behavioral recognition

Since the click sequence contains no spectral structure above 100 Hz, stretching the click sequence is a manipulation that has no impact on the frequency content of the sound in the spectral range of zebra finch syllables (>500 Hz). We found that birds trained to discriminate fast sequences failed to respond above chance levels when the timescale of the clicks was slowed by a factor of two. The slow sequences were truncated at three seconds to match the original stimulus duration.

We next examined whether naive birds could learn to discriminate the slower click sequences if they were exclusively trained on the slower sequences from the outset. Eleven birds were trained in a single-stage training and four birds were trained on the first stage of a two-stage training procedure that is also documented in the Methods section. In contrast to the high success rate for faster click sequences, no bird developed a discrimination ability for the slower click sequences (*Figure 6c*, *Figure 6—figure supplement 2g and h*). The ability to discriminate click stimuli was found only for the faster click sequences.

## Discussion

Songbirds form detailed auditory memories for complex songs, and these memories serve to guide imitative vocal learning (*Nottebohm, 1972*; *Brainard and Doupe, 2002*; *Bolhuis and Gahr, 2006*; *Gardner et al., 2005*). In parallel, a range of songbird species can perform at high levels in operant tasks involving song and synthetic stimulus discrimination (*Gess et al., 2011*; *Sturdy and Weisman, 2006*; *Cynx and Nottebohm, 1992*; *Scharff et al., 1998*; *Stripling et al., 2003*; *Abe and Watanabe, 2011*). While songbird auditory performance has been well documented, the network mechanisms underlying song discriminations have not been studied. In particular, one of the least understood aspects of auditory sequence processing concerns the transformations applied to complex temporal sequences (*Mauk and Buonomano, 2004*).

The present study provides insight into the processing of a simple class of temporal sequences composed of irregularly spaced clicks. We find that after the stimulus passes through the primary thalamorecipient zone — L2a, L2b, and L3 — these temporal sequences are transformed into distinct

population vectors or 'spatial codes'. The mapping of temporal patterns to spatial patterns or ensemble codes provides an opportunity for downstream neurons to perform stimulus discrimination using simple linear classifiers to act on the population vector. For the click stimuli used in this study, reliable discriminations could be made on the basis of the distinct population vectors that arise in L2b and L3, binned in 5 ms time bins.

Operant training revealed that songbirds readily learn to discriminate the Morse-code like click stimuli. The fast-click sequences were behaviorally discriminable with high accuracy for a majority of trained birds. Surprisingly, no animals learned to discriminate click sequences that were slowed by a factor of two, even though secondary auditory areas responded with similar spike rates to the slower stimulus. The slowed sequences evoked inter-peak intervals in primary auditory area PSTH that were longer than the typical intervals between peaks in the PSTH during natural song exposure. We suggest that the ascending auditory pathway in the transition from L2a to L2b and L3 is tuned to process temporal events on the faster timescale (11–40 ms) in a manner that is particularly useful for song memorization and discrimination.

We mention two caveats in the present study. First, the high-pass cutoff frequency of the loudspeakers was 3 kHz. (High frequency tweeters were used for stimulus delivery limiting the spectral content of each click.) We do not know how the spectral content of the click impacts the behavioral discrimination of the slower sequences. In another prior study, zebra finches were able to discriminate sequences of beeps spaced by intervals of up to 300 ms — intervals much longer than those used in our study (*van der Aa et al., 2015*). It is likely that brief clicks and longer tones tap into auditory processing pathways with distinct temporal dynamics, explaining the performance difference. In addition, many details of the temporal discrimination tasks were different in the two studies, and the distinct results may also relate to these task differences. Additional tests will be needed to determine whether or not the spectral content of each click impacts the behavioral performance. Opportunities also exist to further examine the ability of the zebra finch to generalize temporal pattern recognition through time-dilations (*Nagel et al., 2010*).

The second caveat is that the single-unit ensembles studied here were 'virtual ensembles' recorded in different animals; noise correlations within animals could further impact discrimination in ways that were not addressed here (*Zohary et al., 1994*; *Abbott and Dayan, 1999*). While we did not acquire enough high-quality single-unit data to perform the ROC analysis for individual animals, enough units were recorded simultaneously to reveal the transformation qualitatively from primary to secondary responses in summary raster plots (*Figure 3—figure supplements 2* and *3*). These rasters support the view that the sequence transformation described for virtual ensembles will also hold for ensembles of neurons in individual birds.

Much theoretical interest has focused on the question of how brains composed of neurons with short intrinsic timescales can process long-timescale stimuli and generate long-timescale behaviors (*Lashely, 2004*). For temporal stimuli composed of identical units such as clicks, intrinsic cellular or circuit mechanisms must bridge intervals of time from one interval to the next in order to create sequence-specific population responses. To encode the history of the stimulus in the present state of the network, synfire chains, avalanches, or more complex transient dynamics in recurrent networks have all been proposed (*Abeles, 1991*; *Grossberg, 1969*; *Maass et al., 2002*). In other models, persistent currents in single cells bridge intervals of time (*Egorov et al., 2002*). In each of these models, intrinsic dynamics of cortical cells or circuits are used to transfer information about past events into the network responses at a given time.

One effective way of transferring information about prior events into current responses is through feedback connections. The primary auditory area (L2a) in songbirds reportedly receives no feedback from higher-level auditory zones (*Vates et al., 1996*), and the synchronous, low-latency responses in this region may reflect a feed-forward response to thalamic drive. By contrast, all other areas, including the secondary auditory zones examined here (L2b and L3), are more densely interconnected both with each other and with higher-level auditory areas. This anatomical distinction raises the possibility that L2b and L3, but not the primary auditory area, L2a, can sustain reverberant activity that could underlie the temporal sequence transformation observed in L2b and L3. Relevant theoretical constructs for this model include liquid state machine theories (*Maass et al., 2002*). By way of illustration, *Figure 8* reveals the output of a simple reverberant model that recapitulates key features of the observed dynamics. In this case, the model is simply a linear dynamical system driven by click

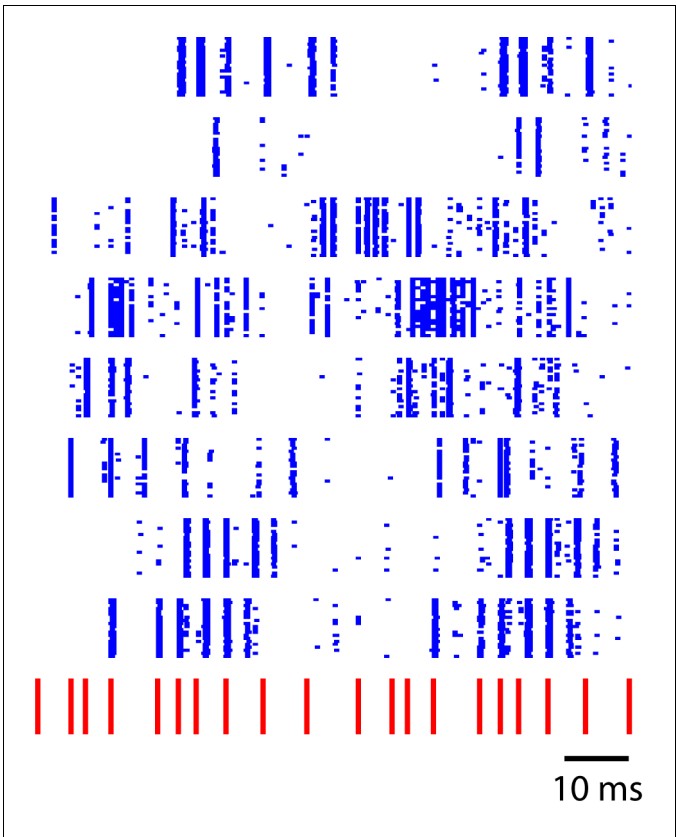

**Figure 8.** Sequence-selective responses in a critically tuned linear dynamical system. Each blue row represents simulated neural responses in a simple linear model. The input stimulus (red) has a temporal pattern similar to the click sequences used in this study. This toy model illustrates a temporal to spatial transformation arising from simple linear dynamics in a recurrent system.

sequences and additive noise, and tick marks represent time points when the vector ($v$) crosses arbitrary threshold amplitude.

$$dv/dt = \alpha M v - \gamma v - \eta \tag{1}$$

In this example, matrix $M$ is a random anti-symmetric matrix, with all imaginary eigenvalues, and $\eta$ is a random noise term. By choosing the time-constants $\alpha$ and $\gamma$ appropriately, the model can produce patterns that resemble spike rasters observed in L2b and L3. *Figure 8*, generated by this linear system, simply illustrates the point that even the simplest recurrent dynamical systems have the capability to transform click-sequence information into distinct population vectors when properly tuned. In this example, the anti-symmetric matrix, $M$, provides a form of 'critical tuning' in which multiple oscillatory timescales are equally excitable, providing richer temporal dynamics than would occur for a typical nonsymmetric random connectivity matrix (*Magnasco et al., 2009*). While the hypothesis that recurrence explains the auditory sequence transformation is appealing, new experimental studies are needed to examine the role of recurrent dynamics in temporal sequence processing in the auditory pallium of songbirds.

While the reverberant models provide an attractive explanation for the sequence transformation observed in L2b and L3, the behavioral discriminations that the birds exhibited here cannot be taken as evidence that supports the reverberant model, strictly speaking. A purely feed-forward counter-hypothesis is that the neurons in secondary auditory areas could demonstrate biophysical integration timescales that solve the sequence discrimination through single-cell properties. To illustrate this hypothesis, we first smoothed the click sequences used for behavior training with three different rectangular windows of timescale $T$ or shorter and built phase-plane traces of these hypothetical

'smoothing units' in 3D space. In this case, three different smoothing windows correspond to hypothetical units with different integration timescales. We then analyzed the minimal time-scale $T$ for which the behaviorally trained sequences could be perfectly segregated in the ROC analysis performed earlier for actual neural sequences. From this analysis, we found that phase plane traces used in the behavioral studies can be perfectly separated if the width of the longest rectangular window was 100 ms or greater. To state this more simply, while the sequences presented to the birds were 3 s long, click rates measured in time bins as short as 100 ms provide a population vector that is adequate for sequence discrimination. This 100 ms timescale cannot rule out either feed-forward single-cell biophysics or recurrent dynamics as contributors to the sequence transformation.

While the circuit mechanisms remain to be established, this study serves to demonstrate both a distinctive transformation of temporal sequences in the transition from L2a to higher-order areas, L2b and L3, and a behavioral capacity of zebra finches to discriminate synthetic click sequences. The transformation of temporal sequences to distinct population vectors may underlie the songbird's advanced discrimination abilities for temporally structured conspecific song.

## Materials and methods

All procedures were approved by the Institutional Animal Care and Use Committee of Boston University.

### In vivo experimental preparation

#### Subjects
For the neural recordings, we examined a total of 11 different adult male zebra finches (*Taeniopygia guttata*).

#### Stimulus
The artificial stimulus set used for electrophysiology consists of nine click sequences with different interval ordering (*Figure 3—figure supplement 1*). Sequence 9 was used only for a subset of operant training tests. The natural song stimulus set consisted of three conspecific songs (n = 13 syllables). During electrophysiological recordings, a neural data acquisition system (RZ-5, Tucker-Davis Technology) triggered a pulse generator to create rectangular pulses (100 µs width) with different intervals, or played the conspecific bird songs. All stimuli were presented in free-field (60~65 dB peak amplitude) over a loudspeaker (bird song) (Companion 2, BOSE Corporation, Framingham, MA, USA) or a tweeter (for clicks) (PLWT4, Pyle Audio, frequency response range: 3–30 kHz).

#### Neural recording
Prior to the electrophysiological recording, the birds were injected with the anti-inflammatory analgesic Meloxicam, via intramuscular injection, and anesthetized (1–2% isoflurane in 0.6–0.8 $\ell$/min O$_2$) for a preparatory surgical procedure to implant a custom-made head-post. Local scalp anesthetic (bupivicaine) was administered subcutaneously (40 µl, 4 mg/kg) and a small (0.18 g) head plate affixed to the skull through light-bonded dental acrylic. This was attached so that the head could later be held at a fixed 55 degree angle during unanesthetized auditory recordings. After recovery from general anesthetic (two hours), the bird was given a booster dose of bupivicaine along the margins of the scalp, placed in a foam restraint, and transferred to a double-walled sound proof chamber (40A-Series, Industrial Acoustics Company, Bronx, NY, USA), facing a loudspeaker or a tweeter. The sound source was located 25 cm away from the bird beak.

We used a four-shank multichannel silicon probe (Impedance: 1–2 MΩ, A4×8–5 mm-50-200-177-A32, Neuronexus, Ann Arbor, MI, USA) to record extracellular spikes. The coordinates for recording were 1.5 mm lateral and 0.8 mm anterior to the bifurcation point of the mid-sagittal sinus. The probe was advanced slowly at the speed of 1–2 µm/s using a motorized manipulator (MP-285, Sutter Instrument Company, Novato, CA, USA) until the tip of electrode was located 1.0–1.6 mm below the surface of the brain. Recordings lasted for 4–5 hr. At the end of the recording, an electrolytic lesion was made at the location of one of the silicon shank tips using a tungsten electrode (10 µA for 10 s). Following this, the birds were deeply anesthetized (110 µl, sodium pentobarbital [250 mg/kg]) and perfused. The extracted brains were stored in 4% paraformaldehyde solution for histology. On the next

day after perfusion, parasagittal 100 μm sections of the brains were prepared (Vibratome Series 1000, Technical Products International, St. Louis, MO, USA) and stained with cresyl violet. Electrode placement was verified by comparing electrolytic lesions to histological landmarks that define the boundaries of Field L (*Fortune and Margoliash, 1992*).

## Spike sorting

To isolate single units, the extracellular voltage traces were high-pass filtered at 500 Hz (third order Butterworth filter) and putative spikes were detected if the voltage traces crossed the positive- and negative-going threshold (*Quiroga et al., 2004*). Then, spikes were re-aligned to the negative peak after resampling up to 250 kHz using the cubic spline interpolation method. Features of the aligned spikes were composed of the first three principal components and wavelet coefficients of spike waveforms (*Quiroga et al., 2004*). A mixture of Gaussian models were fitted to the spike features using an Expectation Maximization (EM) algorithm to build distinctive clusters of spikes with similar spike waveforms (*Pham et al., 2005*). Unit quality was then assessed by signal-to-noise ratio (SNR) and refractory period violations to select well-isolated single units (*Ludwig et al., 2009*). All analyses for spike sorting were performed using custom software written in MATLAB (The Mathworks Inc. Natick, MA, USA).

## Spike-pattern classification

We classified spike patterns into primary and secondary responses on the basis of cross-correlations between spike trains and click-sequence stimuli. The similarity score was defined as the maximum cross-correlation value of normalized PSTH (bin size: 5 ms) with the normalized click stimulus. A unit was classified as primary If the similarity score was above 0.5 and secondary if the score was below 0.5. Physiological classification was validated by histology (*Figure 1b* and *Figure 1—figure supplement 1*), which revealed that although exact coordinates differed in different animals, primary neurons formed a contiguous island within the surrounding zone of secondary-like responses. The continuity and scale of these islands of primary responses were consistent with the known anatomy and location of primary thalamoricipient zone L2a.

## Timescales of neural responses

The timescales of ensemble responses to songs and click sequences in the primary auditory area L2a were characterized by the distribution of intervals between neighboring peaks of the smoothed PSTH (5 ms bin). To smooth the PSTH, we filtered the PSTH with an FIR band-pass filter (Kaiser window, passband: 5–110 Hz, number of coefficient: 2233, sampling rate: 1 kHz, passband ripple is 5% and stopband attenuation is 40 dB). The filtered PSTH was then normalized so that the values were distributed between 0 and 1. Local peaks of normalized PSTH are selected on the basis of three conditions: distance between peak and valley >0.01, peak value >0.3, and peak height relative to the neighboring valley >0.05.

## Phase space trajectory

After dividing responses into two groups (primary or secondary), we built a population vector array that contained all PSTHs of units for different stimuli (bin size: 5 ms). Each vector had $n$ dimensions of data, where $n$ is the number of neurons. To visualize the behavior of multiple neurons (*Figure 4a*), we applied principal component analysis (PCA) on the population vector arrays using functions from MATLAB's Statistical Toolbox.

## Stimulus discriminability

We defined discriminability of neural responses as the minimum Euclidean distance between two different population vector arrays in response to distinct sounds. Before calculating distances, each spike rate trace in a population vector was smoothed with a 30 ms rectangular window. Then, we divided the recording session into two groups (odd vs. even numbered trials) and obtained the distribution of distances in population vector space built from either the same stimulus or across different stimuli. To build the ROC curve, we calculated the true and false positive ratio for discriminating two different stimuli while changing the decision-boundary position.

## Auditory operant-training preparation

Here we describe a method for auditory-operant training that is useful for training zebra finches on challenging discriminations with little shaping procedure. The proposed method uses water reward rather than seed reward (*Picardo et al., 2015*). Zebra finches are adapted to arid conditions and can survive for months in a laboratory setting without access to water (*Cade et al., 1965*), yet they remain highly motivated to work for water. The quantity of water provided in each reward can be as low as 1–5 µl. With this reward quantity, birds work for hundreds or thousands of trials per day.

### Subjects

In the operant task, we trained 53 adult (>90 days post-hatch on the first day of training) male zebra finches (*Taeniopygia guttata*). All birds were housed in the same aviary room and were experimentally naive at the start of training. Once a bird entered the training cage, he remained in the training cage 24 hours a day until the end of training period.

### Food and water

Dehydrated seed (100–110Fº for 12 hr, D-5 Dehydrator, TSM Products) was supplied every two days (seed is dehydrated the day before it is provided in the cage). Soft food (ABBA 97 Ultimate nestling food, ABBASEED) was available once per week. Birds had unlimited access to water on the weekends and every day access to grit. Birds were not exposed to water deprivation conditions prior to training. On a single day of training, birds normally initiated around 800–1300 trials (with a maximum of 4000 trials for one individual). This corresponded to 300–1000 µl of water consumption during training. We provided additional water (0.5–1 ml) after the training if the total volume of water consumption for two days was less than 1 ml. The birds usually drank 0.5–1 ml of water over night when this supplement was provided. In total, through reward and supplement, the experimental birds received an average of 1–1.5 ml of water every day, a number that corresponds to 50% of normal water consumption for zebra finch under certain environmental conditions (*Cade et al., 1965*).

### Operant chamber

In this experiment, 12 identical operant-training cages were used. The training cages (11 inch wide x eight inch high) were kept inside sound attenuation chambers (22 inch wide x 14.5 inch high x 16 inch deep, *Figure 6—figure supplement 1*). All inside surfaces of the chambers were lined with embossed acoustic foam (PROSPEC Composite, Pinta acoustics inc). Inside each training cage, there were two infrared switches (OPB815WZ, OPTEK Technology): one for trial initiation (called the trial port) and one for water reward (called the water port). The water reservoir was located 24 inches above the cage floor and the water valve (EW-01540–02, Cole-Parmer) was placed between the reservoir and spout. The water spout was located in the middle of the infrared switch assembly (water port), so that whenever the bird accessed the water spout, he broke the infrared beam automatically. We used two different sizes of incompressible plastic tubes to make water flow slow enough to create a proper drop size (1–5 µl). An illustration of the tubing is shown in *Figure 6—figure supplement 1*. The sound stimulus was presented through the same tweeter used for the electrophysiology study (PLWT4, Pyle Audio). A microprocessor dedicated to each cage (Arduino Mega 2560, Arduino) controlled stimulus presentation, water delivery, and infrared switches. Individual clicks generated by the Arduino microprocessor were 100 µs long rectangular pulses. Using this microprocessor, the mean jitter in the click interval was 93 µs (data are not shown). Every time the bird tried a new trial, data from the previous trial was transmitted by ethernet to a central data processing computer in the lab and analyzed in real time by a custom made Matlab program (Mathworks, Natick MA, USA). Training ran for five hours per day from Tuesday to Friday each week. The behavior of all birds was monitored through USB webcams in each chamber (Webcam Pro 9000, Logitech).

## Auditory operant-training procedure

In this procedure, a bird can demonstrate learning through two behaviors — by interrupting ongoing playback of a non-rewarding stimulus to reset the trial, or by accessing the water port selectively for rewarding stimuli. We trained birds with two different methods: a two-stage method and a single-stage method. In all experiments, two sounds were presented — a rewarded stimulus (click sequence 2) and a non-rewarded stimulus (click sequence 1 or 9).

### Training procedure during stage 1 of two-stage training

This training starts with only one infrared switch (for trial initiation, on the left side of the cage, *Figure 6—figure supplement 1*). The bird can start a new trial or interrupt playback of the stimulus by breaking the infrared beam any time 200 ms after the start of the stimulus playback. The water spout is on the right side of the cage and the water reward is passively given at the end of the rewarded stimuli, which constitutes 20% of total trials. In this setup, the bird learns to be 'impatient' and to interrupt stimuli that are not followed by reward. In a sense, the bird is 'foraging' for a low-probability rewarded sound. On each day of training, we monitored the latency of trial initiations to two different sequences. During the first one or two days, birds simply explored the training environment and explored the trial port randomly. Gradually, birds realized the existence of passive water reward and started to reinitiate trials earlier on non-rewarding trials than on rewarding trials (right middle panel of *Figure 6—figure supplement 2a, and 2d*. Note the bump of red curve around 5–6 s). In 1–2 weeks of training, birds could re-initiate trials only for non-rewarded trials, and wait for water reward on the rewarded trials.

### Training procedure during stage 2 of two-stage training

Once birds showed significant learning in stage 1, another infrared switch was activated on the water delivery port. Water was no longer delivered passively, but only if the water port was accessed during or just after the playback of the rewarding stimulus. This period, during which reward port access leads to release of water, is called the 'response time-window'. This window was 7 s long from the end of a sequence. If the water port was accessed at any time during non-rewarding trials, or outside of the 7 s response window, a 10 s time-out period ensued, during which the green LED was turned off. Introducing another infrared switch in this stage did not alter the trial reset behavior that was acquired in the first stage of training (*Figure 6—figure supplement 2c and f*).

### Training procedure for single-stage training

In single-stage training, the bird begins training with both infrared switch-contingencies active from the beginning. However, to jumpstart the process, water was also delivered passively at the end of the rewarded stimulus if the bird did not access the water port during playback of the rewarded stimulus. Once the bird learned to initiate trials and encounter water at the water port location, the passive water delivery was shut off. Other than this brief passive delivery period, this method involved no shaping or staging. Birds learned strategies for the use of both infrared switches through exploration (re-initiating trials within 3 s when the non-rewarded pattern was presented and accessing the water port during playback of the rewarding stimulus, *Figure 6a*).

### Operant task behavior evaluation

We used a d-prime measure to estimate the progress of learning:

$$d' = z(H) - z(F) \tag{2}$$

where H is the proportion of correct responses (hit rate) and F is the proportion of incorrect responses (false alarm rate) (*Green and Swets, 1966*).

## Acknowledgements

This work was supported by CELEST, a National Science Foundation Science of Learning Center (NSF OMA-0835976), by the National Institute of Health (NIH R01NS089679), the National Research Council of Science and Technology (NIST) grant from the Korea government (MSIP) (No.CRC-15–04-KIST), and the Ministry of Science, ICT and Future Planning/Institute for Information & communications Technology Promotion as part of the ICT R&D program (R0126-16-1119). We thank Frederic Theunissen, Luke Remage-Healey, Kathy Nordeen, Ofer Tchernichovski, Sarah Bottjer, Elizabeth Regan, and Richard Hanhloser for providing zebra finch song samples from their colonies. We would also like to thank Aniruddh Patel for comments that improved the analysis and manuscript.

## Additional information

### Competing interests

BGS-C: Reviewing editor, *eLife*. The other authors declare that no competing interests exist.

### Funding

| Funder | Grant reference number | Author |
|---|---|---|
| National Research Council of Science and Technology | CRC-15-04-KIST | Yoonseob Lim |
| Ministry of Science, ICT and Future Planning | Institute for Information and Communications Technology Promotion R&D program, R0126-16-1119 | Yoonseob Lim |
| National Science Foundation | NSF OMA-0835976 | Barbara G Shinn-Cunningham Timothy J Gardner |
| National Institutes of Health | NIH R01NS089679 | Timothy J Gardner |

The funders had no role in study design, data collection and interpretation, or the decision to submit the work for publication.

### Author contributions

YL, Designed the study, Collected data, Wrote the paper and analyzed the data; RL, Collected data; BGS-C, Designed the study; TJG, Designed the study, Wrote the paper

### Author ORCIDs

Barbara G Shinn-Cunningham, http://orcid.org/0000-0002-5096-5914
Timothy J Gardner, http://orcid.org/0000-0002-1744-3970

### Ethics

Animal experimentation: This study was performed in strict accordance with the recommendations in the Guide for the Care and Use of Laboratory Animals of the National Institutes of Health. All of the animals were handled according to approved institutional animal care and use committee (IACUC) protocols (Protocol Number: 11-027) of the Boston University, operating under AALAC registration 000197, OLAW assurance A3316-01 and USDA 14-R-0017. All surgery was performed under isoflurane anesthesia, and every effort was made to minimize suffering.

## Additional files

### Supplementary files

• Supplementary file 1. Click-sequence audio files. We provide audio files of all the click sequences used in this study in .wav format. The last number of the file name corresponds to the index of click sequence. For example, Clk_Sequence_1.wav contains audio data for sequence 1.

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
