## [Decision Letter]

Thank you for submitting your article "Transformation of temporal sequences in zebra finch auditory cortex" for consideration by *eLife*. Your article has been reviewed by three peer reviewers, and the evaluation has been overseen by a Reviewing Editor and Timothy Behrens as the Senior Editor. The following individuals involved in review of your submission have agreed to reveal their identity: Shihab Shamma (Reviewer #3).

The reviewers have discussed the reviews with one another and the Reviewing Editor has drafted this decision to help you prepare a revised submission.

Summary:

This manuscript describes experiments using fast and slow click trains to probe the neural mechanisms of temporal sequence discrimination in auditory areas of the zebra finch brain. More specifically, the authors describe a transformation from a temporal code to a spatial code in the higher auditory areas, which may enable birds to discriminate temporal sequences in the temporal range relevant for birdsong.

1) For fast click trains (interclick intervals in the 10 to 40 ms range), there is a transformation of the auditory responses between lower and higher areas. Whereas in the primary auditory area, neurons are highly synchronized, in higher areas the neural responses are less synchronous and evolve through a complex high-dimensional trajectory that carries more information about the sequence of click intervals.

2) For slower click trains (interclick intervals in the 20 to 80 ms range), there is less discrimination of different click sequences in higher auditory areas.

3) At the behavioral level, zebra finches are capable of learning sequences of clicks from the "fast" click trains but not the "slow" click trains. Finally, using and interesting and novel operant conditioning paradigm, the authors show that zebra finches can discriminate fast click sequences but not slow click sequences.

Strengths include the use of click trains to eliminate spectral information and isolate temporal processing, and the integration across multiple levels-primary and higher auditory areas and behavior. The experiments and analysis appear well designed and executed, and the claims of the paper are exciting. However, more thorough and, in some cases, more straightforward analysis of the data is needed, and the clarity and precision of the writing should be improved.

Essential revisions:

1) It seems that the essential claim is that, in the transformation from primary to secondary auditory cortex, the higher area integrates information from the lower areas over some timescale (say 20-40ms) to allow the discrimination of different sequences. The time-constants in the secondary auditory cortex are presumed to be long. This would be consistent with the observation that the slower click sequences (in which fewer of the clicks fall within the integration time) produce less discriminable spike patterns in the higher areas, and are less discriminable behaviorally. It seems that it should be possible to simply measure something akin to integration time-constants of these secondary auditory cells. Such analysis would likely be much more intuitive in the context of the paper, and could supplement or replace the more indirect analyses provided.

2) Similarly, it would be helpful to include a straightforward analysis of the integration window required to discriminate the primary auditory spike trains. It is true that the longer time constant can help conceptually to convert a temporal sequence into a spatial pattern through integration. But it is evident that the time constants in the secondary are not much longer than 2 or 3 clicks. So how would the spatial patters form for a whole sequence? Or is a sequence for these birds just 2 or 3 click patterns? If the birds need, say, 5-10 clicks to discriminate a sequence from another, then there must be some kind of integration that is commensurate with that sequence. Otherwise, the secondary patterns of activity are just a sequence of spatial patterns, and we are still back to the same question: how do we encode a sequence as one spatial pattern? The authors do not address this with respect to their data, since they seem to argue that for these birds, the secondary ACX is sufficient to account for all their behavioral performance. Also, it is not clear why the discriminability is largely eliminated by reducing the click rate by a factor of two. The fast and slow click sequences share click intervals in the range of 20ms to 40ms. And yet these shared intervals do not seem sufficient to yield discriminability in the slow sequence. Why is this? Is it possible that only the shortest click intervals (i.e. between 10ms and 20ms) provide discriminability?

3) A strong argument is made that the inter-peak intervals in the spike trains for their fast click sequences match those observed for natural song, while those for the slow click sequences do not. Given the plots in Figure 2, the reviewers did not find this argument convincing. The interval distribution for natural song appears to be exponentially distributed, while that for the fast click sequence is sharply limited below 40ms. The use of fast and slow click sequences is interesting and important, but the authors should find a less distracting way to motivate and describe their choice of click intervals.

4) The description of what was actually done (particularly regarding physiology) is hard to follow and scattered throughout the paper. When did they use units versus sites, when did they use simultaneous recordings versus pooled recordings, when did (or didn't) they do nested analyses to account for within-bird or within-site variability?

5) The terminology is often misleading; this paper advances our understanding of auditory processing in the bird telencephalon but does not tell us anything about how the cortex works – in fact the majority of the discussion relates to "primary" L2a's lack of recurrent/feedback connections, which is one of the ways in which it is most clearly dissimilar from any mammalian neocortex.

---

## [Author Response]

*Essential revisions:*

*1) It seems that the essential claim is that, in the transformation from primary to secondary auditory cortex, the higher area integrates information from the lower areas over some timescale (say 20-40ms) to allow the discrimination of different sequences. The time-constants in the secondary auditory cortex are presumed to be long. This would be consistent with the observation that the slower click sequences (in which fewer of the clicks fall within the integration time) produce less discriminable spike patterns in the higher areas, and are less discriminable behaviorally. It seems that it should be possible to simply measure something akin to integration time-constants of these secondary auditory cells. Such analysis would likely be much more intuitive in the context of the paper, and could supplement or replace the more indirect analyses provided.*

We agree the hypothesis of longer cellular time-constants was over-emphasized given that there is no direct data such as whole-cell recordings in our study, or in prior studies.

The text has been modified to de-emphasize this unsupported idea in the Introduction and Discussion sections.

We now provide more balance in the Discussion, including equal emphasis on other possible mechanisms, including the presence of recurrent feedback in secondary areas that could extend the effective integration timescale in a way that doesn’t necessary imply a change in membrane biophysics.

*2) Similarly, it would be helpful to include a straightforward analysis of the integration window required to discriminate the primary auditory spike trains. It is true that the longer time constant can help conceptually to convert a temporal sequence into a spatial pattern through integration. But it is evident that the time constants in the secondary are not much longer than 2 or 3 clicks. So how would the spatial patters form for a whole sequence? Or is a sequence for these birds just 2 or 3 click patterns? If the birds need, say, 5-10 clicks to discriminate a sequence from another, then there must be some kind of integration that is commensurate with that sequence. Otherwise, the secondary patterns of activity are just a sequence of spatial patterns, and we are still back to the same question: how do we encode a sequence as one spatial pattern? The authors do not address this with respect to their data, since they seem to argue that for these birds, the secondary ACX is sufficient to account for all their behavioral performance. Also, it is not clear why the discriminability is largely eliminated by reducing the click rate by a factor of two. The fast and slow click sequences share click intervals in the range of 20ms to 40ms. And yet these shared intervals do not seem sufficient to yield discriminability in the slow sequence. Why is this? Is it possible that only the shortest click intervals (i.e. between 10ms and 20ms) provide discriminability?*

This was an important suggestion. We now explain in the manuscript that the sequences used in the behavioral training can be discriminated with smoothing windows on the timescale of 100ms or shorter. As explained in the Discussion, to do this, we first smoothed the click sequences used for behavior training with three different rectangular windows of timescale T or shorter and built phase plane traces of a click sequence in 3D space. In this case, three different smoothing windows correspond to hypothetical units with different integration timescales. We then analyzed in the phase space the minimal time-scale T for which the behaviorally trained sequences could be perfectly segregated. From this analysis, we found that phase plane traces can be separated if the width of rectangular window is greater than 100ms.

Next, we examined whether the neural response over short timescales could discriminate the sequences. (This parallels the ROC analysis we reported for the full 3 second sequence. When applied to just the first 500ms of neural response, the story still held – separation in secondary but not primary areas.) This shorter timescale roughly matches the behavioral response times.

*3) A strong argument is made that the inter-peak intervals in the spike trains for their fast click sequences match those observed for natural song, while those for the slow click sequences do not. Given the plots in Figure 2, the reviewers did not find this argument convincing. The interval distribution for natural song appears to be exponentially distributed, while that for the fast click sequence is sharply limited below 40ms. The use of fast and slow click sequences is interesting and important, but the authors should find a less distracting way to motivate and describe their choice of click intervals.*

This is a good point – birdsong includes many significant behavioral timescales ranging from 20ms to multiple seconds, and the slowed timescale clicks still overlap with many significant timescales in behavior. We keep the observation that the time to space transformation is tuned to a timescale that is common in birdsong, but now show the histograms for the spike sequences for reference only. We no longer argue that the factor of two slowing takes us “out of birdsong” timescale.

*4) The description of what was actually done (particularly regarding physiology) is hard to follow and scattered throughout the paper. When did they use units versus sites, when did they use simultaneous recordings versus pooled recordings, when did (or didn't) they do nested analyses to account for within-bird or within-site variability?*

This has been clarified with a new introduction section to the Results.

In this study, we have recorded single and “sorted” multi-unit responses from primary and secondary auditory areas of 11 different birds. For all the analysis including click sequence discrimination analysis, song response analysis, and PSTH, we used only well sorted single units, and now clarify this in the text.

The only exception is in Figure 3—figure supplement 2: here, we have shown ensembles of all the units recorded in this study. In this figure, we have indicated with an asterix multi-units on the left side of raster that did not pass the high SNR criteria for single units.

Also in the figure, we indicate which recordings came from each bird. All the numbers on the right side of raster correspond to the indices shown in Figure 1—figure supplement 1.

*5) The terminology is often misleading; this paper advances our understanding of auditory processing in the bird telencephalon but does not tell us anything about how the cortex works – in fact the majority of the discussion relates to "primary" L2a's lack of recurrent/feedback connections, which is one of the ways in which it is most clearly dissimilar from any mammalian neocortex.*

Our use of the term cortical is a shorthand that is common in the field, but needed further explanation to simplify, we removed all the terminology that may mislead readers such as cortex and cortical and added terminologies for aviary anatomy.